# Horizontal Transfer of Malignant Traits and the Involvement of Extracellular Vesicles in Metastasis

**DOI:** 10.3390/cells12121566

**Published:** 2023-06-06

**Authors:** Goffredo O. Arena, Stefano Forte, Mohamed Abdouh, Cheryl Vanier, Denis Corbeil, Aurelio Lorico

**Affiliations:** 1Department of Surgery, McGill University, Montréal, QC H3A 0G4, Canada; goffredo.arena@gmail.com; 2Fondazione Istituto G. Giglio, 90015 Cefalù, Italy; 3Mediterranean Institute of Oncology, 95029 Viagrande, Italy; stefano.forte@grupposamed.com; 4Cancer Research Program, Research Institute, McGill University Health Centre, Montréal, QC H3A 0G4, Canada; mohamed.abdouh@mcgill.org; 5Touro University Nevada College of Medicine, Henderson, NV 89014, USA; cvanier@touro.edu; 6Biotechnology Center (BIOTEC) and Center for Molecular and Cellular Bioengineering, Technische Universität Dresden, 01307 Dresden, Germany; denis.corbeil@tu-dresden.de

**Keywords:** cancer, metastasis, nucleoplasmic reticulum, extracellular vesicles, exosomes, oncogene, tumor suppressor gene, microenvironment

## Abstract

Metastases are responsible for the vast majority of cancer deaths, yet most therapeutic efforts have focused on targeting and interrupting tumor growth rather than impairing the metastatic process. Traditionally, cancer metastasis is attributed to the dissemination of neoplastic cells from the primary tumor to distant organs through blood and lymphatic circulation. A thorough understanding of the metastatic process is essential to develop new therapeutic strategies that improve cancer survival. Since Paget’s original description of the “Seed and Soil” hypothesis over a hundred years ago, alternative theories and new players have been proposed. In particular, the role of extracellular vesicles (EVs) released by cancer cells and their uptake by neighboring cells or at distinct anatomical sites has been explored. Here, we will outline and discuss these alternative theories and emphasize the horizontal transfer of EV-associated biomolecules as a possibly major event leading to cell transformation and the induction of metastases. We will also highlight the recently discovered intracellular pathway used by EVs to deliver their cargoes into the nucleus of recipient cells, which is a potential target for novel anti-metastatic strategies.

## 1. Introduction

Metastasis is defined as the spread of cancer cells from the primary site of formation to distant tissues and/or organs. Such spatiotemporal cellular dissemination that occurs through the blood and lymphatic system is considered to be responsible for the vast majority of cancer deaths worldwide [1]. A deeper understanding of metastasis might identify new mechanisms that could be intercepted, leading to new anti-cancer therapeutic strategies. The classic hypothesis of the metastatic process is based on Stephen Paget’s “Seed and Soil” model, according to which the distribution of metastases is not casual, but organ-specific: the “soil”, the proper tissue or organ environment, allows the growth of the “seed”, i.e., certain tumor cells with metastatic potential, owing to the interaction between the cancer cells and the host organ [2]. It was shown that the mechanism by which metastatic cancer cells implant and grow at distant sites involves cross-communication between them and resident stromal/immune cells, via direct physical cell–cell contact or soluble factors, secreted locally or carried systemically by nanosized extracellular membrane vesicles (abbreviated here as EVs) [3,4,5,6,7,8,9,10,11,12].

The role attributed to EVs as mediators of intercellular communication is now emerging in the contexts of embryogenesis, tissue regeneration and the immune system, as well as in various diseases, notably in cancer, where EVs can favor the formation of metastases, as discussed later in this review [13,14,15,16,17]. Different types of EVs have been described, the most common being those released after the fusion of multi-vesicular bodies with the plasma membrane, as exosomes, with a size below 120 nm, or directly released from the plasma membrane of donor cells, as ectosomes/microvesicles (100–1000 nm) [9,18,19,20]. In addition, larger EVs such as apoptotic bodies (1–5 μm), released upon cell fragmentation during apoptotic cell death, and large oncosomes (1–10 μm), released from non-apoptotic membrane blebs of migrating cancer cells harbouring an amoeboid phenotype, have also been described [21,22,23,24,25,26]. Another type of large EVs are migrasomes (0.5–3 μm), released upon the degradation of cell’s retraction fibers left behind by migrating cells or during membrane cell retraction [27,28]. The latter may transport chemokines and cytokines and thus play some role in the dissemination of cancer cells and/or interfere with the immune system [29,30]. In general, EVs contain various types of bioactive molecules, such as proteins, RNAs and/or genomic/mitochondrial DNA, which reflect the genetic status of the donor cells (see below) [31,32,33] and can impact surrounding tissues and/or distant organs [34]. In the latter case, it remains to be determined how circulating EVs reach them, but an interesting study has demonstrated that cancer EVs can breach the intact blood–brain barrier via transcytosis [35] (reviewed in Ref. [36]).

Once released into a given bodily fluid, EVs can encounter other cells and transfer their contents to them. In this process, the membrane constituents of EVs and those associated with the plasma membrane of recipient cells could determine the selective targeting to a particular recipient cell type and the mechanism of uptake or internalization (reviewed in Refs. [20,37]). The uptake of EVs, notably those derived from cancer cells, can play a major role in the metastatic process. First, through hemo-lymphatic circulation, EVs could reach distant sites, where they can stimulate non-neoplastic stromal/immune cells to support tumor growth, thus creating the appropriate “soil” of the pre-metastatic niche (PMN) [4,5,38,39]. Thus, EVs could induce neo-angiogenesis [40,41,42] and immunosuppression [43,44,45]. In addition to this important contribution to PMN, EVs may play a role in metastatic organotropism [5], i.e., the affinity of each cancer type to metastasize to specific organs [46,47] (see below).

The impact of EVs on a given physiological process is modulated by their content. In cancer, especially in metastasis, certain proteins associated with the surface of EVs contribute to cell invasion. As examples, the highly glycosylated form of the extracellular matrix metalloproteinase inducer (EMMPRIN), present at high levels in EVs from metastatic breast cancer patients, contributes to tumor invasion in the surrounding tissue [48]. Similarly, the tetraspanin protein CD9, also highly expressed on EVs, is thought to be associated with cancer cell invasiveness by promoting EV internalization, which stimulates cellular transformation (reviewed in Ref. [49]), while some integrin proteins expressed on the surface of EVs allow their selective uptake by organ-specific cells, leading to PMN preparation [5]. In addition to protein content, various types of RNAs were reported in EVs. Among them were messenger RNA (mRNA), microRNA (miRNA) and long non-coding RNA (lncRNA). The latter may induce permanent changes in chromatin structure and the regulation of gene expression [50,51], acting as inducers of pro-metastatic transformation [51,52,53,54,55]. Cancer cell-derived apoptotic bodies and large oncosomes may also carry large fragments of DNA [21,22,56]. Even smaller EVs such as exosomes have been shown to contain large fragments (up to 10 kb) of double-strand DNA, and carry mutations of parental cancer cells [57,58,59]. They can also carry mutated DNA fragments, possibly harboring cancer driver mutations [57,60]. EVs isolated from the serum of tumor-bearing mice were found to contain the DNA, which reflects the genetic status of tumor-donor cells, including the amplification of the oncogene *c-Myc* as well as retrotransposon transcripts [61]. Together with other nucleic acids, transposable elements can be transferred to normal cells, indicating a vast repertoire of genetic information available for horizontal gene transfer. While it has been reported that EVs can mediate the transfer of chromatin-associated double-stranded DNA and the mutated H-ras oncogene [62], it is important to stress that their uptake and cellular impact are strongly determined by host cells, with mesenchymal cells being the most receptive targets leading to transient transformed phenotypes [63]. This interesting study also suggests that while oncogenic EVs carrying a single mutated oncogene (RAS) are part of an important regulatory and communication mechanism, they are not capable of inducing a permanent horizontal and genetic transformation, suggesting the potential for the latter and/or that pre-existing alterations in recipient cells are required (see below).

The universality of the “Seed and Soil” model has been challenged over the years [64,65,66,67]. For instance, James Ewing proposed that the patterns of tumor metastases could be attributed to the anatomy of vascular and lymphatic drainage from the primary tumor [68]. Accordingly, Ewing’s hypothesis maintains that tumor cells follow the circulatory route, draining from the primary tumor, and stop non-specifically in the first organ encountered. Although there is evidence to support this theory, it cannot be opposed to the “Seed and Soil” model; the two are not mutually exclusive and may depend on the tissue origin and cancer types. Note that the Ewing hypothesis can account for the distribution of circulating EVs, which do not have active mobility, as described for migrating cells (see below), and are driven by circulatory flows. Decades later, alternative theories of metastasis formation were elaborated, such as metastasis by cellular fusion, genometastasis and, more recently, the horizontal transfer of malignant traits (HTMT) [69,70,71]. In this review, we will outline the key steps of these models of metastatic dissemination, with a special emphasis on the potential involvement of EVs in such processes. As EVs may be limited, especially when acting over a long distance from donor cells, we will also highlight a new intracellular transport pathway allowing EV cargoes to efficiently encounter their targets and/or reach the nuclear compartment of recipient cells, potentially allowing the genetic or epigenetic modification of recipient cells and their transformation into malignant cells [72,73].

## 2. Model of Primary Tumor Cell Migration and Growth: The “Seed and Soil” Model

At least five major sequential steps can be identified in the “Seed and Soil” model, stemming from the original Paget’s hypothesis, which allows the cancer cells to metastasize to an anatomic site where the local microenvironment is favorable, just like a seed will only grow if it lands on fertile soil, owing to a concurrent action of circulating cancer cells and microenvironmental cues at the secondary sites [2]. They are: (a) the detachment of cancer cells from the primary tumor and the invasion of the surrounding tissues; (b) intravasation into the bloodstream or lymphatic system; (c) survival and migration through the circulation; (d) arrest and/or trapping in capillary and extravasation into a given distant organ; and finally (e) the seeding and colonization of secondary sites. For more details on all these steps, see these two excellent reviews [74,75]. Several lines of evidence suggest that, while cancer cells can detach from the primary tumor and trans-migrate into the circulatory system, the actual colonization of remote sites depends on their ability to reach, survive and proliferate efficiently in distant organs [3]. Several studies have confirmed this hypothesis, such as the observation that when syngeneic mice were implanted with small lung fragments in ectopic locations, melanoma cells metastasized to both normal lung and ectopically placed lung, but not to any other tissues [76]. In another study, the adhesion of tumor cells to the microvascular endothelium of their respective target organ was responsible for the localization of metastases [77]. In some cases, tumor cells undergo dormancy in secondary organs, and can be found in the G_0_ phase of the cell cycle, or cannot stimulate the angiogenesis required for their growth [78].

Organotropism supports the idea that the colonization of circulating cancer cells depends on local conditions at remote sites, rather than solely on passive diffusion or random distribution [46,79]. Various chemokine- and growth-factor-mediated mechanisms have been described to explain tumor organotropism (reviewed in Ref. [80]). A classic example is the CXCR4–CXCL12 chemotaxis axis [81]. CXCR4, a chemokine receptor expressed by most cancer types, including cancers of epithelial, mesenchymal and hematopoietic origin [82], has a critical role in cell migration and metastasis to organs that secrete its ligand, CXCL12, also known as stromal cell-derived factor-1 (SDF-1) [83]. CXCR4 overexpression is associated with poor prognosis in many types of cancer [84]. Interestingly, melanoma EVs were found to induce melanoma cell osteotropism by activating the SDF-1/CXCR4/CXCR7 axis, where CXCR7 is required by melanoma cells to promote their chemotaxis toward SDF-1 gradients [85]. Cell polarization, migration and chemical gradient sensing as different steps of chemotaxis have been reviewed elsewhere [86,87].

The PMN is also induced by signals released by primary tumors into the general circulation that promote molecular and cellular changes in the microenvironment, enabling circulating cancer cells to seed and give rise to metastatic lesions. In a landmark publication [88], Kaplan and colleagues demonstrated that a specific sub-population of bone marrow-derived hematopoietic progenitor cells expressing vascular endothelial growth factor receptor 1 (VEGFR1) and VLA-4 (i.e., integrin α4β1) moved to tumor-specific PMN, where they formed cellular clusters before the arrival of tumor cells; in this case, Lewis lung carcinoma cells. Furthermore, cancer cell-derived growth factors upregulated one of the VLA-4 ligands, namely, extracellular matrix fibronectin, in resident fibroblasts, thereby providing a permissive niche for incoming tumor cells. Such an observation demonstrated that the expression patterns of fibronectin and VEGFR1^+^VLA-4^+^ clusters dictate organ-specific tumor spread. In the same study, the authors showed that pro-metastatic events occurred in multiple organs when conditioned media collected from the B16-F10 metastatic melanoma cell line were used [88]. The latter observation is consistent with the wider range of metastatic sites observed in melanoma patients [5]. More generally, the studies of the temporal evolution of PMN have revealed that early events, i.e., those that precede the establishment of the tumor mass at the metastatic site, include clot formation, vascular disruption and increased permeability (reviewed in Ref. [3]). Vascular disruption is associated with an augmented metastatic burden, as demonstrated in murine models [4,89]. The hallmarks of vascular integrity loss are the establishment of hyperpermeability, the acquisition of an aberrant cellular morphology in the vascular endothelium and the promotion of breaks in vascular basement membrane. In breast cancer and melanoma models, the role of factors regulating the activity of angiopoietins has been associated with the loss of blood vessel integrity and augmented permeability. For instance, breast cancer cells release transforming growth factor beta (TGFβ) that modulates the expression of angiopoietin-like 4, thus enhancing the permeability of lung microvessels [90]. Similarly, melanoma cells induce an upregulation of angioprotein 2 and matrix metalloproteases (MMP) 3 and 10 in the lungs before cancer cell colonization [91]. Other soluble factors, such as cyclooxygenase 2, epiregulin, MMP1 and MMP2, increase vascular permeability in breast cancer models, with effects evident in both the primary tumor and distant organs’ vessels [92].

Vascular alterations are then followed by changes in resident cells and the recruitment of monocytes and metastasis-associated macrophages that, in a positive feedback loop, promote the further extravasation and survival of metastasizing cancer cells [93,94]. In addition to these phenomena, extracellular matrix (ECM) proteins, such as tenascin C and periostin, can be produced by tumor cells themselves or by tumor-associated cells and play a pleiotropic role in metastasis progression by promoting invasive cell behavior, cancer migration and growth at the metastatic sites and neo-angiogenesis and cancer cell viability under stress [95,96]. These molecular and cellular processes seem to confirm that cancer-induced factors are important cues in determining the metastatic process.

## 3. Role of EV-Mediated Intercellular Communication in the “Seed and Soil” Model

EVs released by cancer cells are one of the unexpected factors that could determine and influence the cellular transformation, vascular permeability and the establishment of PMNs, thus preparing the “soil” in the target organs for metastasis (reviewed in Refs. [97,98]). As mentioned above, and regardless of their subcellular origins, EVs and their cargoes play various roles in intercellular communication under physiological conditions, as well as in the pathogenesis of many diseases [10]. Interestingly, the total number of EVs in blood plasma increases in cancer, indirectly highlighting their potential contribution to cancer progression and metastasis [99,100,101,102]. Indeed, specific cancer-associated microenvironmental factors [103], such as decreased pH (acidity) due to cancer cell metabolism [104,105,106] or hypoxia [107], have an impact on increased EV release and the subsequent contribution to malignant tumor phenotypes. Low pH conditions could also have an effect on EV uptake, notably in fusion with host cells [104].

In metastases, integrin molecules associated with EVs can determine the organotropism [5,108]. The interaction of EVs with cell-surface-associated ECM in a given tissue/organ may allow their specific retention and uptake by resident cells at the predicted metastatic destination, activating intracellular pathways [5,109]. For examples, EVs harboring integrins α6β1 and α6β4 were associated with lung metastasis, while those with integrin αvβ5 were associated with liver metastasis due to preferential fusion with different types of resident cells at their predicted destination, lung fibroblasts and liver Kupffer cells, respectively [5]. Interestingly, the targeting of the specific integrins resulted in decreased EV uptake at the level of the organs site of metastasis [5]. The integrins (e.g., αvβ6) were also shown to be transferred from prostate cancer cells to recipient cells via exosomes and remained active in the host cells, where the αvβ6-mediated signaling pathway could modify the tumor microenvironment [110,111,112,113]. Avβ3-expressing EVs were reported to promote tumor growth and induce neuro-endocrine differentiation in recipient cancer cells [114]. The integrins (e.g., β1) carried by EVs can also promote the anchorage-independent growth of tumor cells, possibly through the induction of changes in the composition of EVs secreted from cancer cells [110,111].

Hence, through EVs, cancer cells communicate between themselves and with cancer-associated fibroblasts (CAFs), or with other stromal cells, such as mesenchymal stromal cells (MSCs), vascular endothelial cells and surrounding immune cells, to promote their own growth and spreading [97,115,116]. In the context of an intercellular communication mechanism between cancer cells, Schillaci and colleagues showed that metastatic SW620 colon carcinoma cells were able to transfer, via EVs, an aggressive amoeboid phenotype to isogenic non-metastatic SW480 cells (Figure 1, see legend for detail) and to induce endothelial permeability, while non-metastatic cell-derived EVs were unable to induce such a transformation [117]. The mechanism underlying the observed cellular transformation has not yet been fully pinpointed [118], but certain differentially represented proteins in EVs derived from metastatic cancer cells have been suggested as potential candidates for the stimulation of pro-metastatic features. For example, thrombin, which is more abundant in EVs released by metastatic colon cancer cells versus non-metastatic cells, can promote the mesenchymal-to-amoeboid transition in recipient cancer cells as well as the monolayer alteration and loss of cell–cell contacts in human umbilical vein endothelial cells, both effects mediated by the activation of the RhoA/ROCK pathway [117]. This does not exclude a role of nucleic acids carried by EVs in the cellular transformation and endothelial permeability (see below). Interestingly, an ameboid-like phenotype has also been associated with cannibalistic behavior in metastatic cells [119], suggesting that both may be induced by EVs. MSC-derived EVs containing cell cycle inhibitory microRNA can induce cancer cell dormancy [120]; however, due to the rapid cannibalism of MSCs, the contribution of microRNA transfer to tumor dormancy has been reported as minimal [121]. A CD9-dependent invasion of cancer cells into MSCs, with phenomena of entosis (cell-in-cell), was also reported by Rappa and colleagues [122].

Regardless of the molecular mechanism, we have shown that the cancer cell transformation can be inhibited by the application of a Fab fragment derived from the anti-CD9 antibody, which prevents the uptake of EVs [123,124] (Figure 1A). Similarly, EVs containing annexin A6 released from pancreatic CAFs induced aggressiveness in pancreatic cancer [125], and CAFs were found to stimulate the migration ability of scirrhous-type gastric cancer cells [126]. In the latter cases, the tetraspanin CD9 protein was also found to be a critical factor in EV uptake.

**Figure 1 cells-12-01566-f001:**
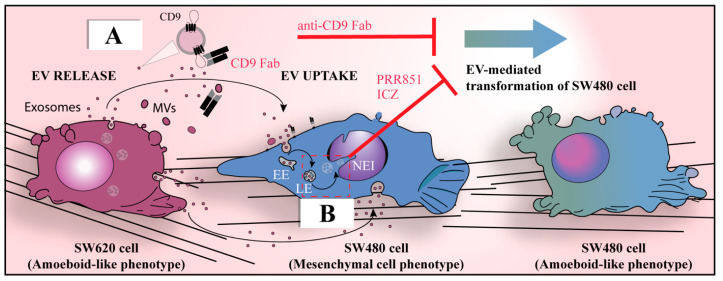
EVs derived from metastatic colon cancer cells promote the morphological transformation of non-aggressive cancer cells. (**A**,**B**) The established isogenic colorectal cancer cell lines consisting of highly metastatic SW620 cells and non-metastatic SW480 cells provide a cellular model in which EVs (exosomes and/or microvesicles (MVs)) derived from the former can transform the latter and impact their malignant properties, notably by changing their migration patterns from mesenchymal to amoeboid motility [127,128]. The latter phenotypic alteration can be prevented either by intercepting CD9^+^ EVs derived from SW620 cells using a Fab fragment antibody directed against CD9 that blocks their internalization by SW480 cells (**A**), or by blocking the nuclear transfer of cargo carried by endocytosed EVs using drugs such as itraconazole (ICZ) or PRR851 (**B**), which act on the tripartite protein complex required for the translocation of late endosomes into the NR (for more details, see Section 10). EE, early endosome; LE, late endosome; Fab, fragment antigen-binding; NEI, nuclear envelope invagination. Modified from Ref. [124].

Cancer cells can also prime non-neoplastic cells via their EVs to facilitate cancer growth and metastasis through various cargoes. Evidence that EVs can mediate the transfer of cancer traits to normal MSCs derived from the same tissue/organ was provided by an in vitro study on colorectal cancer [129]. EVs isolated from cultured primary or metastatic cancer cell lines were able to induce morphological and functional changes in colonic MSCs. These changes included the formation of atypical microvilli and pseudopods, and an increase in EV release and augmented proliferation, migration and invasion rate. This evidence confirmed a clear involvement of EVs in phenotypic transformation at a local level, paving the way for the possibility of the induction of phenotypical transformation at a distance. Likewise, EVs derived from highly metastatic melanomas that contain high levels of the receptor tyrosine kinase c-met increased the number of distant melanoma metastases (e.g., in bone and lung) compared with those released from a poorly metastatic melanoma cell line [4]; the mechanism advocated consisted of the influence or “education” of bone marrow progenitor cells through the horizontal transfer of the EV-associated phosphorylated c-met to PMNs and/or an increase in vascularization [4]. These findings were nonetheless questioned, as a reproducibility study was unable to detect phosphorylated c-met in EVs, and the reversal of an EV-induced increase in metastatic burden by EVs with reduced c-met expression was not statistically significant [130]. Nevertheless, the induction of the formation of PMN by cancer EVs administered to mice prior to tumor cell injection has been reported in various other studies [5,6,12,42,131,132,133]. As examples, glioblastoma-derived EVs stimulated cerebral microvascular endothelial cells, resulting in increased angiogenesis [42] and pancreatic ductal adenocarcinoma (PDAC)-derived EVs inducing Kupffer cells to secrete transforming growth factor β (TGFβ) and hepatic stellate cells to upregulate fibronectin production in the liver, resulting in an increased metastatic burden [6]. This fibrotic microenvironment enhanced the recruitment of bone marrow-derived macrophages. Interestingly, in the same study macrophage migration inhibitory factor (MIF) was found to be highly expressed in PDAC-derived EVs, and its inhibition prevented liver PMN formation and metastasis. To further define the mechanisms, the authors conducted a gene expression analysis of Kupffer cells exposed to PDAC EVs and showed an increase in transcripts involved in liver fibrosis pathways, specifically genes associated with soluble growth factors (e.g., CTGF, EDN, IGF, PDGF and TGFβ) [6]. Along the same lines, increased levels of TGFβ in patients with pancreatic cancers were associated with disease progression and poor prognosis [134]. The pro-metastatic circuit coordinated by EVs from PDAC cells may represent the molecular substrate of such a condition and provide a functional explanation for the efficacy of anti-TGFβ compounds in preclinical and clinical studies [135,136]. Finally, it is important to mention that EVs derived from, for example, breast cancer cells, can impact vascular endothelial barriers by targeting the tight junction protein ZO-1 through EV-associated miRNA-105, and thus facilitating the metastasis process [5,6,12,131,132,133].

Collectively, these selected datasets highlight the contribution of cancer EVs in the modulation of various pathways associated with the metastatic process and suggest that they may play a central role in the transmission of messages to metastatic sites to promote the “homing” of circulating cancer cells. Thus, EVs have added another level of complexity to the Seed and Soil model, which may in some ways address some of the limitations of this classical hypothesis based solely on cancer cell migration and the appropriate metastatic niche.

## 4. Limitations and Challenges of the Seed and Soil Model

As mentioned in the Introduction, the universal validity of the original Seed and Soil model has, nonetheless, been questioned over the years due to certain shortcomings or a lack of experimental data supporting the entire underlying process of metastases [64,65,66,67]. These shortcomings include (i) the low number of tumor cells that can actually circulate in the blood, e.g., less than 0.1% cells remain viable and <0.01% surviving circulating cells can produce metastasis [137,138]; (ii) the lack of definitive evidence that a single cancer cell is capable of sequentially performing all steps of the metastatic process, i.e., separation from the primary tumor, intravasation, survival in the circulation, extravasation and successful colonization [139]; (iii) the contrast between the concept of tumor dormancy, i.e., a prolonged latent state of asymptomatic micro-metastatic disease prior to overt metastasis formation [140], and uncontrolled proliferation [141], considered one of the main hallmarks of cancer; (iv) the poor correlation between bone marrow micro-metastases and their clinical manifestation [142]; and (v) differential gene expression profiles between primary cancer and metastatic cells [143].

Furthermore, the passage of cancer cells from the circulation to the metastatic site involves the ability to overcome physical constraints determined, for instance, by the presence of the tight capillaries’ junctions. This capability entails the reacquisition of new features such as reduced actin cytoskeleton anisotropy, cell stiffness and focal adhesion density. Moreover, once in the bloodstream, circulating tumor cells must acquire the ability to survive a variety of stresses, such as avoiding anoikis (a form of apoptosis due to the loss of integrin-dependent anchorage to the ECM), evading the immune system, and overcoming hemodynamic shear forces [144]. The acquisition of these abilities, according to the conventional model, might be determined by the expression of epithelial-to-mesenchymal transition (EMT) transcription factors such as Snail family transcriptional repressor (Snail), Twist, Zinc finger E-box binding homeobox 1 (Zeb1) and different miRNAs, as well as epigenetic and post-translational regulators. Upon reaching the metastatic site, the same cell would need to undergo a reversal of state, i.e., mesenchymal-to-epithelial transition (MET), to be able to home in the metastatic niche (reviewed in Refs. [145,146,147]). Although cancer cells have shown the ability to convert between the epithelial and mesenchymal phenotypes in vitro, it is yet to be proven that an epithelial cancer cell is able to alternate between the two states in vivo. Currently, and to our knowledge, there is no convincing evidence that circulating tumor cells have the ability to revert fully to an epithelial state in vivo [148,149] (reviewed in Refs. [150,151,152]). As mentioned in these review reports, the EMT/MET model can concur with other models such as collective migration, making it difficult to grasp the definitive mechanism.

The strongest element supporting the classical model of metastasis is the immunohistochemical similarity between primary cancer cells and the metastatic deposits. However, microarray analyses have shown that the gene expression patterns of primary breast tumors differ from those of their respective lymph node metastases, and that there are genes that are characteristically different in metastatic cells compared with their counterpart in primary tumors [143,153,154,155]. Moreover, the genomic and expression heterogeneity of disseminated tumor cells in the bone marrow of early breast cancer patients has been observed [156]. A possible explanation of the phenotypic differences could be the malignant transformation of bone marrow cells rather than the migration of breast cancer cells from primary sites. These discrepancies strengthen the notion that immunohistochemical similarity does not necessarily imply a common origin and that metastatic cells do not always derive from the migration and replication of the particular primary malignant clone.

## 5. Metastasis by Cellular Fusion

Fusion between cells is involved in many physiological processes, including fertilization and myogenesis, among others [157]. Such cellular processes could also occur in cancer, and explain their progression and metastases. As proposed by the physician Otto Aichel more than a century ago, metastasis could result from the fusion of cancer cells with healthy cells (e.g., leukocytes), resulting in hybrid cells that retain the properties of both parent cells [158] (reviewed in Refs. [159,160,161]). One of the first supports for the fusion came from a set of experiments by Barski and Cornefert, who mixed two distinct tumor cell lines and co-injected them into host mice, producing hybrid clones that, upon extraction and reinjection, gave rise to tumors in secondary recipient animals [162]. Similarly, Goldenberg and colleagues demonstrated a relationship between cell fusion and metastasis by injecting a female human astrocytic glioma into the cheek pouch of male golden hamsters, where lethal metastases developed rapidly with the formation of human–hamster hybrids, as revealed by chromosomal analyses [69]. They also reported that human ovarian adenocarcinoma cells grafted into nude mice induced the transformation of adjacent normal stromal cells into murine sarcomas [163]. They concluded that the concept of cancer cells transferring malignant properties to adjacent normal cells needed to be reconsidered as a mechanism involved in the evolution of mixed tumors or heterogeneous cell populations, as shown by cell–cell fusion and the segregation of human chromosomes and/or of transforming “genes”, resulting in the horizontal transmission of malignancy. Following these pioneering studies, the fusion of cancer cells with other cancer cells, MSCs or leukocytes at the primary tumor or at metastatic sites has been extensively documented (reviewed in Refs. [161,164]).

More recently, one of our groups reported that aggressive breast cancer cells MDA-MB-231 (hereafter MDA) and MA-11 spontaneously fused with MSCs, while less aggressive MCF-7 or benign mammary epithelial cells did not [122,165]. Hybrids showed predominantly mesenchymal morphological characteristics. An analysis of single-nucleotide polymorphisms revealed genetic contributions from both parental partners to hybrid tumors and metastasis. Both MDA and MA-11 hybrids were tumorigenic in immunodeficient mice, and some MDA hybrids had an increased metastatic capacity compared to parental MDA [165]. Both in culture and as xenografts, hybrids underwent DNA ploidy reduction and reversal to a breast carcinoma-like morphology, while maintaining a mixed breast cancer–mesenchymal expression profile. The fusogenic activity of CD9 was shown to be essential for the fusion of MDA or MA-11 cells [122]. Interestingly, CD9 knockdown inhibited not only cell–cell fusion but also tumor growth and metastasis in immunodeficient hosts [122], suggesting a direct relationship between fusogenicity and the invasiveness of breast cancer cells. In such a context, it is possible that cell–cell fusion is mediated by CD9^+^ plasma membrane protrusions emerging from cancer cells [122]. The observation that CD9^+^ EVs released from egg cells are essential for sperm–egg fusion also suggests a potential role for EVs to promote cell–cell fusion in the context of cancer metastasis (see references in [49]).

Altogether, these studies unequivocally demonstrate that cell–cell fusion events do occur in human cancers; moreover, tumor-normal hybrid cells were detected in metastases, supporting the hypothesis that cell–cell fusion could explain the enhanced malignant behavior of cancer cells and suggesting that this mechanism might play a role in the ability of a cancer cell to become metastatic.

## 6. The Genometastatic Hypothesis

In 1905, Ehrlich and Apolant reported the sudden appearance of sarcomatous tissue within a mouse mammary tumor with the replacement of carcinoma tissue with pure spindle-cell tumors [166]. These observations were followed several decades later by the finding mentioned above, among others, where the subcutaneous implantation of human ovarian carcinoma cells grafted into nude mice resulted in two cancer populations, a human carcinoma and a murine sarcoma, suggesting the possibility of the transfer of factors from the human cancer cells to murine mesenchymal cells that would induce malignant transformation [163]. Remarkably, only the murine sarcoma cells were metastatic in mice, indicating the transfer of full neoplastic traits to the benign murine mesenchymal cells. The induction of stromal tumors in host animals after the xenotransplantation of human epithelial cancers was confirmed by other studies, and it was observed that the mechanism, although unclear, resembled a viral-like transfer [167,168,169].

Following the reports that the horizontal transfer of genes in simpler organisms such as bacteria and fungi induced antibiotic resistance and adaptation to new environments [170,171], Holmgren and colleagues raised the question of whether DNA could be transferred from one somatic cell to another via the phagocytosis of apoptotic bodies [172]. They showed that the cultivation of apoptotic bodies derived from Epstein–Barr Virus (EBV)-carrying cell lines with either fibroblasts, monocytes or endothelial cells resulted in the uptake of DNA and the expression of EBV-specific markers in the recipient cells. The expression of the EBV markers, Epstein–Barr nuclear antigen 1 as well as EBV-encoded small RNAs, could be detected up to five weeks after the beginning of co-cultivation experiments. They concluded that DNA may be recovered from apoptotic bodies and reused by somatic cells, and further speculated that similar mechanisms of horizontal DNA transfer might be important under conditions characterized by high levels of apoptosis, as observed in tumors treated with chemotherapy or irradiation [172].

Several examples of cancer transformation putatively induced by the horizontal transfer of DNA have been reported in the literature. In 1999, García-Olmo and colleagues found that, a few weeks after the subcutaneous injection in rats of cancer cells stably transfected with a plasmid expressing the prokaryotic gene chloramphenicol acetyl transferase (CAT) under the control of the cytomegalovirus promoter, CAT was detected in their plasma [71]. More surprisingly, upon the injection of plasma taken from CAT-tumor bearing rats into the peritoneal cavity of non-tumor-bearing rats, CAT was found in extracts of lungs, kidney, spleen and liver, suggesting the transfer of the prokaryotic gene to cells located at a distance [71]. Subsequently, they demonstrated that plasma collected from K-ras-mutated colon cancer patients was able to oncogenically transform NIH-3T3 cells, rendering them capable of forming undifferentiated carcinomas when injected subcutaneously in non-obese, diabetic-severe combined immunodeficiency (NOD-SCID) mice [173]. Human-mutated K-ras sequences were detected in the transformed subcutaneous masses and in samples of healthy tissues harvested from the liver, lungs and plasma of the injected animals [173]. The authors concluded that metastases might develop as a result of the transfer of oncogenes, derived from primary tumors and present in the circulating plasma, to susceptible cells in distant target organs. The term “genometastasis” was then used to describe this phenomenon, challenging for the first time the current dogma in which the metastatic process is based on cell migration [71] (reviewed in Ref. [174]). These findings were later confirmed by the observation of the capability of cell-free extracellular DNA to induce cell transformation and tumorigenesis by treating NIH-3T3 recipient cells with serum derived from colon cancer patients or the supernatant of SW480 human cancer cells [175]. NIH-3T3 cells are a widely used murine cell line derived from embryonic fibroblasts and immortalized through repeated transfers at low inoculation densities [176]. Interestingly, the authors noted that cell transformation and tumorigenesis of recipient cells did not occur if either serum or supernatants derived from SW480 cells were depleted of DNA. These data are consistent with a previous study demonstrating that the SW480 cell supernatant was capable of transforming NHI-3T3 cells, which acquired mutated human K-ras [177]. In addition to these in vitro data, they demonstrated that horizontal cancer progression mediated by circulating DNA occurs in an in vivo model where immunocompetent rats subjected to colon carcinogenesis with 1, 2-dimethylhydrazine had an increased rate of colonic tumors when injected in the dorsum with human SW480 cells as a source of circulating oncogenic DNA, which could be neutralized by treating these animals with DNAse I/protease treatments [175]. Thus, they concluded that cancer cells emit into the circulation systems biologically active DNA to foster tumor progression.

The genometastatic hypothesis has left several questions unanswered. How could the cell-free genetic materials circulate and be resistant to degradation in the circulation? What is responsible for oncogenic transformation, free DNA or DNA contained in EVs? How does cell-free DNA enter cells, notably their nuclei, and integrate? What are the characteristics of recipient cells susceptible to oncogenic transformation? Under what conditions would these recipient cells be oncogenically transformed by circulating DNA? Is there a correlation between the ability of patients’ plasma to transform NIH-3T3 cells and the prognosis of the respective colon cancer plasma donors? Until now, none of these experiments had been performed on human cells as recipients and, more importantly, none of the murine cells transformed into cancer cells had been able to achieve a phenotypic differentiation resembling the cancerous histotype of the plasma donors.

## 7. Horizontal Transfer of Malignant Traits Model

About fifteen years after the conceptualization of the genometastasis hypothesis, in a series of reports we reported the malignant transformation of non-malignant human cells into cancer cells by exposure to serum or serum-derived EVs from cancer patients, introducing the model of HTMT as an alternative way to explain the dynamics of cancer metastasis [70,153,178,179,180].

The HTMT model concurs with the genometastatic hypothesis that the metastatic process is not due to the migration of cancer cells per se through the general circulation, as postulated in the Seed and Soil model, but to the transfer of genetic material. The novelty of HTMT is based on the following five observations and postulates: (A) oncogenic information is carried by small EVs circulating in the blood of patients even at the precancerous stages, i.e., when tumor cells have not yet reached the stage of full malignant transformation [179,181,182]; (B) the mutation of at least one oncosuppressor gene in recipient cells in the PMN induces the expression of proteins that facilitate the uptake of cancer EVs [70,180]; (C) the phenotypic changes induced in recipient cells depend on the type of cell that uptakes the EVs rather than the source of the oncogenic message [179,183]; (D) the ability to evade the immune system can be transferred from the primary cancer to distant cells through cancer EVs [153,179]; and (E) metachronous metastases could be due to the nuclear integration of key EV-derived cancer-related genes that are expressed after a period of latency [153].

### 7.1. HTMT and Preneoplastic Lesions

The potential of EVs circulating in the blood of patients in precancerous stages to promote malignant transformation has been suggested experimentally. For instance, EVs isolated from the sera of patients with dysplastic lesions or carcinomas in situ, which by definition are pre-cancerous lesions made of cells that have not yet acquired the ability to metastasize, caused malignant transformation in immortalized human HEK-293 embryonic kidney cells, which have an insertion of approximately 4.5 kb adenoviral DNA in chromosome 19 [22]. Upon transplant into NOD-SCID mice, transformed HEK-293 gave rise to tumors whose histology was compatible with poorly differentiated carcinomas with a high mitotic index [181], suggesting that cancer EVs circulate in the blood before the full neoplastic transformation is complete and cell invasion has occurred. Furthermore, such data reinforce the concept brought forward by the genometastatic and HTMT models that the metastatic process might be partially or completely independent of cell migration [153].

The discovery that, according to the HTMT model, EVs circulating in patients with neoplastic disorders, ranging from dysplastic to metastatic lesions, have a transforming ability on oncosuppressor-KO cells (see below), is the basis of a cancer screening test that can distinguish healthy patients from patients with cancer or at risk of developing cancer [181,182,183]. This ability displayed by oncosuppressor-mutated cells to be transformed by cancer EVs has been incorporated in a unique biological platform for cancer screening called MATERD (Metastatic And Transforming Elements Released Discovery platform) [181,182,183]. Among the peculiarities of this system is its intrinsic ability to detect neoplastic disorders even in patients who had only dysplastic lesions or cancers in situ, and its ability to detect cancer factors even years after the resection of the primary tumor to predict metastatic recurrence. In this regard, MATERD sensitivity is higher than the latest liquid biopsy tests based on the detection of circulating cancer cells, circulating DNA and circulating miRNA [184,185,186]. The clinical results obtained with the MATERD test confirm that (i) cancer factors circulate in the blood prior to the complete cancerous transformation of the cells, undermining the concept that cell circulation is paramount to metastasis formation in distant organs [181,182,183], and (ii) cancer factors are still found circulating in the serum after primary cancer has been removed and for several years afterwards, therefore strengthening the concept that these factors do not require the presence of cancer cells to induce metastatic disease [181,182,183].

### 7.2. Oncosuppressor Genes As Gatekeepers

From a histopathological viewpoint, cells must acquire a series of mutations and transition through the phases of metaplasia, anaplasia and dysplasia before becoming fully cancerous [180]. Therefore, HTMT would occur at the metastatic site where local cells harbor mutations that predispose them to cancerous phenotypes by impairing DNA repair and cell cycle control and/or by regulating the uptake of oncogenic factors from the extracellular milieu [179]. A key distinction between the Seed and Soil model and the genometastatic hypothesis/HTMT is that in the revisited concepts of the Seed and Soil model, cells in the PMN would be expected to respond to signals from the tumor cells by altering the nutrients, immune response, and ECM in a way that facilitates the arrival and subsequent establishment of circulating tumor cells [187]. Instead, the genometastatic hypothesis/HTMT predicts that cells located in the PMN of metastatic organs will respond to circulating signals released from tumor cells located at primary sites by acquiring a malignant phenotype if they have an “initiation” event such as an existing oncogenic mutation [70,153,178,179].

For example, when immortalized cell lines such as human HEK293 kidney embryonic cells, PNT-2 prostate cells [188], which exhibit alterations in cell cycle control, MCF10 cells carrying a mutated oncosuppressor gene, like phosphate and tensin homolog (*PTEN*), or the human fibroblast cell line carrying a CRISPR-Cas9-based deletion of the tumor suppressor breast cancer gene 1 (*BRCA1*) are exposed to blood serum of cancer patients or EVs derived therefrom, they transform into malignant cells which, upon subcutaneous injection into immunosuppressed mice, give rise to cancer masses [178,180,181,182,183] (Figure 2). In contrast, unaltered cells such as human embryonic stem cells, adult liver fibroblasts and MSCs do not [178]. Furthermore, the deletion of *BRCA1* in human fibroblasts significantly increased the uptake of cancer EVs compared to wild-type fibroblasts, suggesting that this specific genomic alteration has an impact on their biological characteristics [70]. The BRCA1 protein is responsible for repairing damaged DNA [189]. Mass spectrometry analyses of *BRCA1*-KO fibroblasts’ plasma membranes revealed that *BRCA1*-KO induces the de novo expression or overexpression of plasma membrane receptors, adhesion proteins or associated proteins such as integrins (α4/β6), carcinoembryonic antigen-related cell adhesion molecules (CEACAM), epithelial cellular adhesion molecule (EPCAM), E-cadherin (CDH1), galectins 3 and 4 and AFADIN, among others [180]. The pharmacological blockage of these newly expressed cell surface proteins dramatically decreased the percentage of cells that internalized EVs, and consequently inhibited cancer formation in mice, corroborating the hypothesis that oncosuppressor genes indirectly influence the uptake of oncogenic factors associated with cancer EVs [180]. It might be more than a coincidence that many of the above-mentioned overexpressed proteins are involved in the metastatic process and facilitate the “homing” of the circulating cancer cells to distant organs, as postulated in the “Seed and Soil” model. In the HTMT model, these proteins allow the preferential uptake of cancer EVs in cells located in distant organs. The malignant EV cargo, once delivered into the cell, would hijack the replicative machinery inducing the malignant transformation at the “metastatic” sites.

### 7.3. The Malignant Metastatic Phenotype Is Dependent on the Cell Type That Uptakes the Cancer EV-Derived Oncogenic Message

Another important distinction between the Seed and Soil and the HTMT models is the role of the PMN in the development of overt metastases. When *BRCA1*-KO fibroblasts (see above) were exposed to different types of cancer sera, regardless of their origin (e.g., breast cancer, lung cancer or lymphoma), they always acquired a malignant epithelial phenotype compatible with a lower gastrointestinal tract differentiation (CK7^−^/CK20^+^/CDX2^+^), whereas *PTEN*-deficient MCF10A cells exposed to the same patients’ sera consistently acquired a malignant phenotype suggestive of upper gastrointestinal tract differentiation (CK7^+^/CK20^−^/CDX2^−^) [183]. According to the HTMT model, the occurrence of two types of primary cancer in the same patient (30% of cancer cases) would be secondary to the uptake of the same oncogenic message by two different cell types and its subsequent expression at different points in time rather than being caused by chronic exposure to the same risk factors [183,190,191,192]. As a proof of this concept, the injection of murine colon adenocarcinoma-derived EVs into the blood stream of immunodeficient NOD-SCID mice resulted in the formation of adenocarcinomas in the lung, with characteristics of both poorly differentiated colon cancer (positive for CK20, CDX2 and AE1/AE3) and lung cancer (positive for Thyroid transcription factor 1 (TTF1) and Napsin) (Figure 3), confirming that EVs derived from a given cancer type target different cell populations at the metastatic niche and induce their transformation [183].

### 7.4. EV-Mediated Interactions of Cancer Cells with the Immune System

Both the Seed and Soil and the HTMT models require that the immune system fails to recognize developing metastatic tumors. While no mechanism of immune tolerance has been suggested by the proponents of the genometastatic model, we recommend the excellent review by Kerkar and Restifo for the mechanisms of immune tolerance compatible with the Seed and Soil model [193]. According to the HTMT hypothesis, the transfer of HLA proteins is particularly important in the context of immune tolerance; in fact, whole-genome sequencing of DNA isolated from *BRCA-1* KO fibroblasts (see above), before and after their transformation in colon cancer cells by colon cancer EVs, showed that gene variants codifying for the extra-membrane portion of HLA proteins were transferred through EVs [153,179]. The finding that mutated cancer genes that codify for HLA proteins closely involved in immunological recognition can be shuttled through EVs to transforming cells suggests that the ability to escape the immune system is one of the malignant traits that can be transferred horizontally.

### 7.5. Cancer Dormancy According to the HTMT Model

Cancer cell dormancy postulates that, although uncontrolled proliferation is a distinct feature of cancer cells, they can stop proliferating and hibernate for long periods until reactivated by unknown signals [140,194,195]. The HTMT model, involving an interplay of cancer EVs, immune cells, and other supportive cells in determining the course of the neoplastic disease, is consistent with the concept of cancer cell dormancy (Figure 4). In fact, it theorizes that cancer EVs released by the primary carcinoma travel through the lymphatics to the regional nodes, where they interact with lymphoid cells [196]. The interplay between EVs and lymphoid cells determines either the destruction of the cancer genetic material contained in EVs or immune tolerance to these oncofactors. If tolerance ensues, EVs can freely travel through the bloodstream, undetected by the immune system, and reach a PMN. There, the cancer genes integrate in the genome of the cell and might be either expressed, determining malignant transformation of the cell, or remain silent and become activated later in time, determining the phenomenon of latency (Figure 4). This hypothesis is similar to the viral dormancy, where viruses such as Varicella-Zoster or Herpes Simplex can establish a life-long latent infection once integrated into the genome of the host cell, punctuated by periods of virus recrudescence, following a decline of the T cell-mediated immunity [197]. A lack of control from the immune system would cause the reactivation of the integrated viral genes with active replication of the virus and cell infection, even decades after integration in the nervous ganglia [197]. This concept that lymph node metastasis might be a hallmark of generalized disease rather than a physical reservoir of cancer cells has gained strength due to the clinical evidence that lymph node dissection and their complete surgical evacuation has little to no impact in several types of cancers [198,199,200,201].

The specific cargoes of cancer cell-derived EVs that would cause such effects at the tumor-draining lymph node level, as well as in distant cells, and the intracellular pathway(s) involved in recipient/target cells still remain poorly understood [202]. As mentioned above, several independent studies have reported the presence of DNA in EVs, and subsequent investigations have established that EV-associated DNA (e.g., BCR/ABL DNA) can reach the recipient cell’s nucleus, causing, for example, chronic myeloid leukemia in immunodeficient mice [203]. In addition to DNA carried by EVs that can integrate into genomic DNA, it has also been shown that epigenetic changes induced by non-DNA EV cargo in the genome of recipient cells may result in malignant transformation transmissible to the cell progeny [204]. Thus, cancer dormancy may be at least theoretically explained by the integration of cancer cell genes or epigenetic changes in EV-primed cells and their subsequent expression at a later time due to a failure of not-yet-defined homeostatic mechanisms, possibly involving the immune system [153] (Figure 4). The finding that EVs contain retrotransposons capable of insertional mutagenesis supports this possibility [42]. Alternatively, cancer EVs themselves could exist in a protective tissue environment and, when induced by an unknown signal, be internalized and cause cell transformation.

### 7.6. Molecular Mechanisms Involved in the HTMT Model

The nature of the molecular mechanisms responsible for the HTMT process has not yet been clarified. The whole genomic sequencing and transcriptome analysis of cancer EVs and the recipient *BRCA1*-KO fibroblasts prior to and after exposure to cancer EVs confirmed that the active transfer of nucleic acids notably mutated cancer genes and their active transcription [179,205]. Xeno-transplants of colon cancer EV-transformed *BRCA1*-KO fibroblasts displayed the epithelial colorectal cancer phenotype, indicating MET. This in vivo evidence correlates at a molecular level with the differential expression of genes involved in the MET process, as well as with cell growth and cell death at both transcript and protein expression levels [179]. Thus, a decrease in the mesenchymal markers Snail1, Snail2, Zeb1, Zeb2, N-cadherin (CDH2) and fibronectin was observed, while the expression of the epithelial marker CDH1 was increased. In cancer EV-exposed *BRCA1*-KO fibroblasts, the expression of the cell cycle progression inhibitor CDKN1A and the cell death inducer mouse double minute 2 (MDM2) was decreased, whereas the expression of the oncogenes MYC and HRAS and the antiapoptotic factor BCL2L1 was increased [179]. These data led to the conclusion that colon cancer mRNAs transferred through EVs were able to modulate the expression of transcription factors that induce a change in the fate of the *BRCA1*-KO fibroblasts through the activation of the MET pathway and the regulation of cell growth and apoptosis [179]. Of note, colon cancer miRNAs, transferred through EVs to the *BRCA1*-KO fibroblasts, were also involved in the MET induction and subsequent acquisition of metastatic features [179].

## 8. Unanswered Questions Raised by the HTMT Model

Although the laboratory evidence gathered so far about the HTMT model has been based on cancer-derived small EVs, whether exposure to large oncosomes or cancer-derived apoptotic bodies could obtain the same effect needs to be investigated. Additionally, while it has been described that the same cancer EVs induce different cancer phenotypes depending on the type of target cell (epithelial vs. mesenchymal), whether the type of oncosuppressor mutation might also have a role in the expression of the phenotype needs to be ascertained. Most importantly, the EV cargo and its molecular mechanisms that trigger the malignant transformation need to be defined. So far, the experiments performed to explain the HTMT dynamics have confirmed the presence of cancer-derived DNA, mRNA and miRNA in the transformed cells. Which EV-contained genes and/or epigenetic mechanisms cause the malignant transformation needs to be established. Little is known about the true composition of the malignant cargo that, once it has been taken up into the cells, is able to perform such dramatic changes. It cannot be ruled out that distinct EVs and their cargoes act synergistically when they are taken up. The evidence that circulating free DNA alone can induce undifferentiated malignant transformation triggers questions regarding the role that other genetic moieties and entities, transferred through the EVs, might have in the process [188]. The malignant transformation of a healthy cell might be secondary to a process similar to the induction of pluripotency in a skin cell induced by the simple epigenetic activation of only four genes [206]. A proper understanding of the composition of the EVs’ cargo and the function of each component is, in our opinion, the path to dissect out in order to clarify the mechanics behind the HTMT model and its possible role in metastatic disease.

## 9. A Novel Intracellular Pathway Accounting for the Nuclear Transport of EV Cargo

In the modified Seed and Soil and HTMT models, the fate of EVs after internalization into recipient cells should have a major impact on their functional role, but unfortunately little is known about this, including about their subcellular distribution in the cytoplasmic compartment and how their cargoes reach a specific molecular target. After endocytosis, the most common cellular entry route for EVs, internalized EVs traffic through endosomal compartments, where the fusion of late endosomes with lysosomes causes the degradation of EV-derived components (see Ref. [20] and references therein). In this recognized and accepted pathway, the EV cargo would not elicit a cellular response. Alternatively, the acidic environment of late endosomes may promote EV membrane fusion with the limiting endosomal membrane, allowing soluble EV cargoes to reach the cytoplasm, and perhaps their targets [207]. Such an exciting scenario remains difficult to conceive, unless there is a massive uptake of EVs, as EV cargoes contain limited amounts of biomaterials. The latter phenomenon might not occur in the context of cancer and metastasis, as intercellular communication between cancer donor cells and acceptor cells at the PNM might rely on a small fraction of cancer cells at the metastatic site and/or over a long distance between primary and secondary sites.

In search of a model that could address this limitation, we recently described a novel nuclear pathway that delivers EVs to the nucleoplasmic reticulum (NR), composed of invaginations of the inner (type I) or both inner and outer nuclear membrane (type II) [208]. This novel pathway consists of a fraction of Rab7^+^ late endosomes carrying endocytosed EVs that translocate by a microtubule-dependent mechanism into the NR, specifically to type II nuclear envelope invaginations (NEIs) [72,73] (Figure 5A). Through interactions of the outer nuclear membrane-associated vesicle-associated membrane protein-associated protein A (VAP-A), cytoplasmic oxysterol-binding protein (OSBP)-related protein-3 (ORP3) and small GTPase Rab7, a tripartite protein complex (named VOR, an acronym for the three proteins involved) allowed the docking of late endosomes onto the outer nuclear membrane and their translocation into NEIs. After the fusion of EVs and the endosomal membrane, the discharge of EV cargoes such as soluble proteins and nucleic acids into the cytoplasmic core of NEIs could allow their concentration therein and/or their translocation into the nucleoplasm through the local nuclear pores [72,73] (Figure 5B). Thus, the NEI may play a dual role in the transfer of EV information, either by increasing the probability that EV cargoes (proteins and mRNAs) will meet or bind their host targets as newly synthesized mRNAs exported out of the nucleus, or by facilitating the nuclear import of EV cargoes. The entry of EV cargoes (proteins and nucleic acids) into the nucleus has been observed in numerous studies, by ourselves and others [72,73,209,210]. Of note, chromosomal DNA sequences in EVs from cardiomyocytes were shown to be transferred into the nuclei of target fibroblasts [211].

The VOR complex appears to be the main pathway by which EV cargo enters the NR, because their nuclear transfer was almost completely prevented in the absence of VAP-A or ORP3 [72]. As demonstrated with the SW480/SW620 cell model (see Figure 1), the phenotypic and functional pro-metastatic transformation of SW480 cells by SW620 cell-derived EVs were impeded following VAP-A or ORP3 silencing [212]. These studies support the view that the VOR complex is part of the novel pathway used by cancer-cell-derived EVs to transfer their content to the nucleus of recipient cells, suggesting a potential mechanism to recruit them to foster metastatic growth.

## 10. VOR Complex Inhibition As a Means to Intercept the EV-Mediated Nuclear Transfer of Oncogenic Factors

The disruption of EV-mediated bidirectional communication between cancer cells themselves at the primary tumor site and/or non-neoplastic stromal/immune cells at the metastatic site may be a novel approach to inhibit cellular transformation and PMN development. There are currently no anti-cancer drugs known to target communication between cancer-cell-derived EVs and the nuclear compartment of host cells, in particular the nuclear transfer of EV cargoes. Our groups hypothesized that compounds that interfere with protein interactions of the VOR complex would prevent such nuclear transfer of EV cargoes, and thereby inhibit the downstream pathways leading to the cellular transformation [212]. In search for such inhibitors, we came across FDA-approved antifungal drug itraconazole (ICZ) that had previously displayed an inhibitory activity on enterovirus and hepatitis C virus replication through binding to OSBP and ORP4, members of the same family to which ORP3 belongs [213]. We found that ICZ disrupts the binding of Rab7 to ORP3-VAP-A complexes, and consequently the translocation of late endosomes into NR, resulting in the inhibition of EV-mediated pro-metastatic morphological changes including the cell migration behavior of colon cancer cells (Figure 1B) [212]. With novel, smaller triazole derivative drugs (e.g., PRR851), the inhibition of the VOR complex was maintained, although the ICZ moieties responsible for antifungal activity and interference with intracellular cholesterol distribution were removed [212]. Knowing that cancer cells hijack their microenvironment and that EVs derived from them determine the PMN, small-sized inhibitors of nuclear transfer of the EV cargo into host cells could find cancer therapeutic applications, particularly in combination with the direct targeting of cancer cells.

Altogether, whether the Seed and Soil or HTMT model (or both) is prevalent in the development of metastasis in which EVs emerge as important players promoting cellular transformation that facilitates the metastatic process, intercepting EV-based intercellular communication may be an innovative approach to limit cancer propagation.

## 11. Conclusions

The different roles attributed to EVs in many pathological conditions, including cancer, are revolutionizing our understanding of the metastatic process, among other things. Their function in the development of the PMN and recent data on the EV-induced malignant transformation of oncosuppressor-KO cells deserve further investigation. Targeting the uptake of EVs and the intracellular pathways they use upon internalization could lead to new therapies that interfere with metastasis. Interestingly, the word “metastasis”, originally from Greek, appeared in the English language in the late 16th century as a rhetorical term, implying ‘rapid transition from one point to another’, while in a medical context, metastasis simply means the transference of the seat of disease [214,215]. Therefore, its original meaning may still encompass all possible mechanistic explanations for the “spreading” of the cancer disease from the original site to distant organs.

## Figures and Tables

**Figure 2 cells-12-01566-f002:**
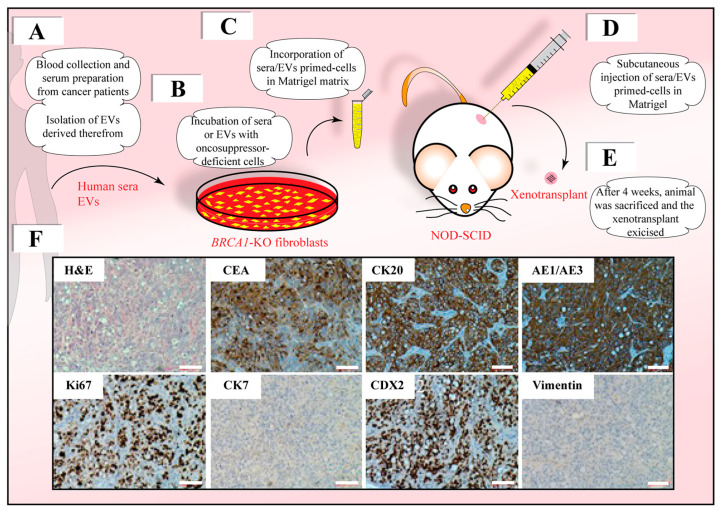
Target cells efficiently internalize EVs derived from colorectal cancer patient sera, and upon inoculation into mice give rise to tumors displaying an intestinal adenocarcinoma phenotype. (**A**–**E**) Sera prepared from patients with colorectal cancer and liver metastasis or EVs isolated from them (**A**) as described in Ref. [70] were incubated with oncosuppressor-deficient cells such as *BRCA1*-KO fibroblasts for a period of two weeks (**B**). Afterwards, cells were harvested and incorporated in Matrigel matrix (**C**) prior to their subcutaneous injection into NOD-SCID mice (**D**). Four weeks after cell transplantation, the animals were sacrificed and the xenotransplant excised (**E**). (**F**) Tissues containing growing tumor lesions were processed for Hematoxylin and Eosin (H&E staining) and/or immunohistochemistry for various markers, as indicated. Note that fibroblasts changed their fate (i.e., loss of Vimentin), and expressed markers of highly proliferative intestinal adenocarcinoma (Ki67, carcinoembryonic antigen (CEA), cytokeratin (CK)20, homeobox transcription factor CDX-2, and anion exchanger (AE)1/AE3), while being negative for the epithelial marker CK7. Micrographs were similar to those presented in Ref. [180]. Scale bar: 100 μm.

**Figure 3 cells-12-01566-f003:**
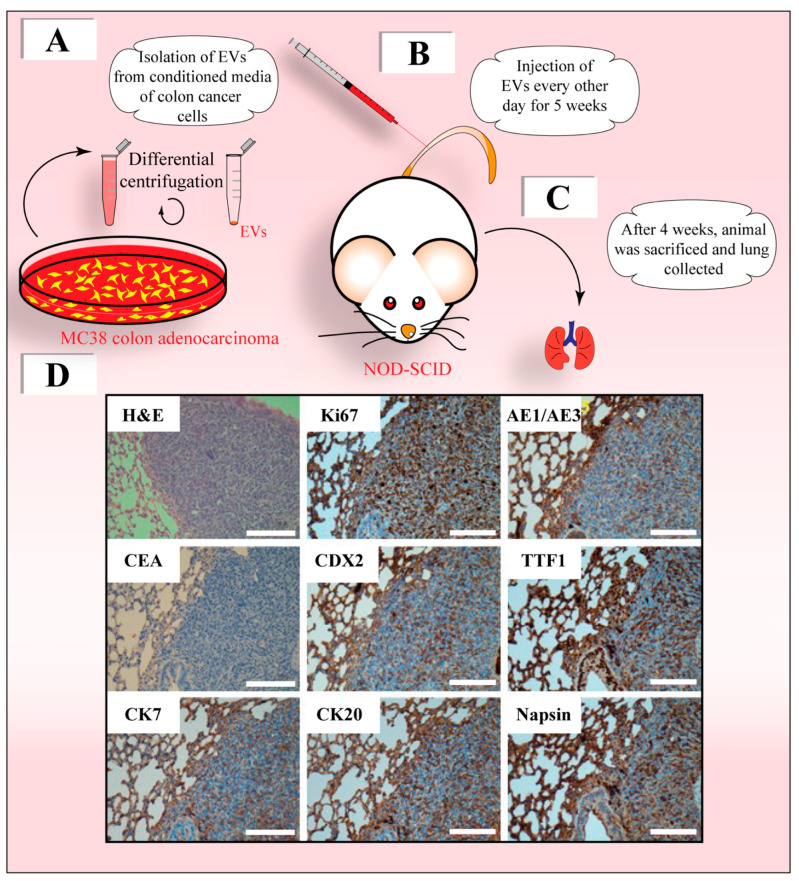
Cancer-cell-derived EVs transfer malignant traits in vivo. (**A**–**D**) EVs (about 4 × 10^6^) enriched by differential centrifugation from the conditioned medium of murine MC38 colon adenocarcinoma cells (**A**, for technical details see Ref. [179]) were injected in the lateral tail vein of NOD-SCID mice every other day for 5 weeks (**B**). Four weeks later, the animals were sacrificed and lungs were collected (**C**). Tissues were processed for H&E staining and/or immunohistochemistry for various markers, as indicated (**D**). The tumors that developed had a proliferative phenotype, as indicated by Ki67 labeling, and had characteristics of poorly differentiated colon cancer, as they were positive for CK20, CDX2 and AE1/AE3 markers, and of lung cancer, as marked by CK7, TTF1 and Napsin. Micrographs were similar to those presented in Ref. [179]. Scale bars: 50 µm.

**Figure 4 cells-12-01566-f004:**
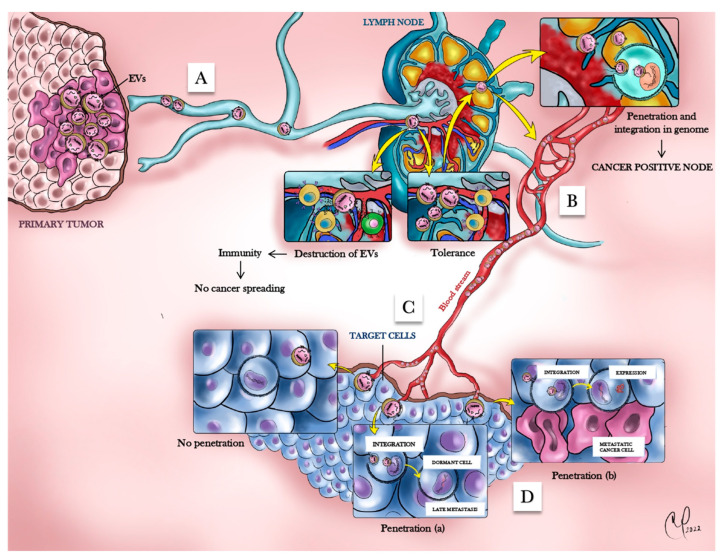
Potential roles of EVs and their cargo in the horizontal transfer of metastatic traits in vivo. (**A**) The primary tumor cells release factors, either naked or encapsulated in EVs, that travel through the lymphatic system into the regional lymph nodes. (**B**) In the lymph nodes, the development of immunity may scavenge these factors with the subsequent inhibition of the metastatic process. Alternatively, tolerance might ensue with subsequent uptake of cancer factors and integration within lymph node cells. (**C**) If oncotolerance is established, oncofactors can freely travel through the bloodstream, undetected by the immune system, and reach a PMN. (**D**) At the PMN, three distinct scenarios can occur: 1. resident cells may be refractory to the uptake of the oncogenic factors (no penetration); 2. receptive putative target cells may express specific receptors or others proteins that facilitate oncofactors’ entry/penetration and integration into the genome without activation of gene transcription (penetration (a)); or 3. oncofactors may be taken up by target cells, with subsequent integration and immediate expression of cancer genes that would determine synchronous metastasis (penetration (b)). In the case of penetration (a), a subsequent decline in immune function would lead to the reactivation of the integrated genes with malignant transformation of the dormant cell and initiation of late metastasis. Adapted from Ref. [153]. The illustration is not drawn to scale.

**Figure 5 cells-12-01566-f005:**
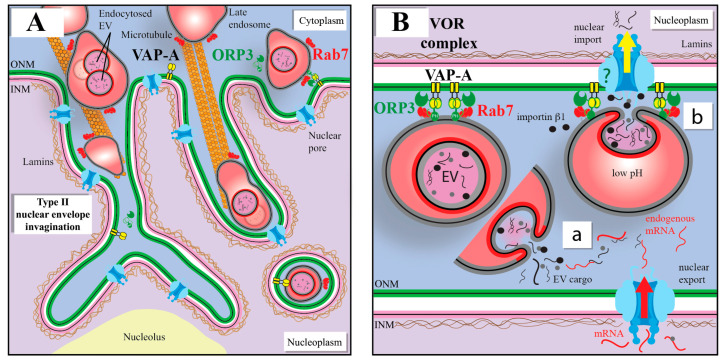
Representation of nuclear envelope invagination-associated late endosomes containing endocytosed EVs and their implications for EV-mediated signaling. (**A**) After endocytosis, EVs move into the endosomal system, and in Rab7^+^ late endosomes they are translocated by a microtubule-dependent mechanism into the NR, especially into type II NEIs, which are often found in close proximity to the nucleolus. Nuclear translocation of late endosomes containing endocytosed EVs requires the interaction of VOR complex proteins: outer nuclear membrane (ONM)-associated VAP-A (yellow), lipid-binding ORP3 (green) and endosome-associated Rab7 (red). A lamin-rich proteinaceous meshwork underlies the inner nuclear membrane (INM), and nuclear pore complexes are found in type II NEIs [72,73]. (**B**) Late endosomes containing endocytosed EVs tether to the ONM via the VOR complex in which cytoplasmic ORP3 binds through its FFAT motif (FFAT being an acronym for two phenylalanines (FF) in an acidic tract) to the membrane protein VAP-A. The pleckstrin homology domain (PH) of ORP3 may mediate its binding to the late endosomal membrane. The domain interaction of Rab7 with the VAP-A-ORP3 complex needs to be defined. After the EV late endosomes fusion, which may be stimulated by a low pH environment, their cargoes are released into the cytoplasmic core of the NEI. Such a restricted space could promote interactions between EV cargoes (e.g., proteins or miRNAs) and their host targets, especially those exported from the nucleus (e.g., mRNA) (a). Alternatively, potential docking (?) of late endosomes to nuclear pores via the VOR complex could further promote the nuclear import of EV cargoes in which importin β1, including that carried by EVs, could play a role, as suggested by importazole treatment [72,73] (b). Modified from Ref. [20].

## Data Availability

Data sharing is not applicable to this article as no datasets were generated or analyzed during the current study.

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
