# Peer review of "Horizontal Transfer of Malignant Traits and the Involvement of Extracellular Vesicles in Metastasis"

_cells, 2023, doi:10.3390/cells12121566_

Round 1
Reviewer 1 Report
This review article is very well written, well organized, comprehensive and on a timely topic. It offers new ideas and critical evaluation of the literature.
It will be heavily quoted and is very appropriate for the readership of Cells.
The role of extracellular vesicles (EVs) released by cancer cells and their uptake is well discussed and recent discoveries on cargo delivery into the nucleus of recipient cells are reviewed.
A minor text revision is recommended:
•Page 5: alphaVbeta6, not alphaV6beta6
•Two additional references are recommended: Krishn et al, JEV 2020; Quaglia et al, JEV 2020: both papers refer to the role of EVs in the cross-talk between cancer cells and the neighboring cells.
1. Quaglia, F., Krishn, S.R., Sarker, S., Pippa, R., Domingo-Domenech, J., Kumar, G., Fortina, P., Liu, Q., Languino, L.R. Small extracellular vesicles modulated by αVβ3 integrin induce neuroendocrine differentiation in recipient cancer cells, Journal of Extracellular Vesicles (2020); 9(1):1761072. DOI: 10.1080/20013078.2020.1761072. PMCID: PMC7448905
2. The role of extracellular vesicles in cancer.
Cell. 2023 Apr 13;186(8):1610-1626. doi: 10.1016/j.cell.2023.03.010.PMID: 37059067 Review.
Author Response
We are pleased that the Reviewer appreciated our Review. We have corrected the name of αvβ6 integrin and cited the three references proposed.
Reviewer 2 Report
Horizontal transfer of malignant traits and the involvement of extracellular vesicles in metastasis by Goffredo O. Arena et al
Of course the role of extracellular vesicles in tumor progression and metastasis is a central issue in experimental and clinical oncology and this a central issue of this review as well.
The authors have discussed some scientific reports on this issue basing their discussion on both the Paget's "Seed and Soil" hypothesis and more recent hypothesis more focused on the role of extracellular vesicles. However they entirely disregarded some crucial evidences that, probably more than others, support their hypothesis.
First of all the authors do not quote and discuss probably the first evidence that tumor exosomes may directly transfer the cancer traits to normal mesenchymal stem cells deriving from the same tissue/organ. This issue together with the well known ability of tumor exosomes to set up a pre-metastatic niche has been discussed by a international group of authors.
There are clinical studies clearly showing that the levels of plasmatic exosomes are significantly higher in tumor patients than in healthy individuals and the number of plasmatic exosomes was proposed as a real tumor marker, possibly useful in cancer patients follow-up.
By the same token the authors do not discuss the role of acidic microenvironment in increasing the release of exosomes by cancer cells, that very often correlates with the number of plasmatic exosomes and with the expression of tumor markers and surrogate tumor markers on the same exosomes.
Another very important issue is the importance of cannibalic behaviour in metastatic cells that is also associated with the ameboid-like phenotype of cancer cells that the authors discuss in their review. In fact, this phenotype has been described in tumor cells deriving from metastatic lesions while being absent in cells from primary tumors. It is therefore highly conceivable that it might be “transferred” from the circulating tumor exosomes.
In general the authors should use exosomes instead of extracellular vesicles, inasmuch as the vast majority of the studies report data on exosomes
the english is acceptable
Author Response
We thank the Reviewer for his/her comments and for recognizing that the topic of the review is clearly of importance. We have added to the text paragraphs and references in regard to the 4 points raised by the Reviewer, namely the cross-talk between cancer exosomes and mesenchymal stem cells, the importance of the acidic microenvironment, the phenomenon of cannibalism and the increased number of exosomes in blood plasma of cancer patients. The new paragraphs are highlighted in yellow and are on pages 5-6 and 17.
Regarding the preference for exosomes vs. extracellular vesicles, we do agree that many of the cited papers mention exosomes, but because the current technology does not allow to clearly distinguish exosomes from ectosomes and other types of extracellular vesicles, we have preferred to maintain the name of extracellular vesicles throughout the text, as also recommended by the Minimal Information for Extracellular Vesicles Committee (MISEV).
Reviewer 3 Report
The review is detailed, the hypotheses of the metastatic process are comprehensively considered. Factual errors were not found.
The only remark - Figure 4. The text is blurry under the letter "C".
Author Response
We appreciate the positive remarks of the Reviewer. We have improved the presentation of Fig. 4, as requested.